# A synergistic strategy to develop photostable and bright dyes with long Stokes shift for nanoscopy

Gangwei Jiang[1,6], Tian-Bing Ren [1,6], Elisa D'Este[2], Mengyi Xiong[1], Bin Xiong [1], Kai Johnsson [3,4], Xiao-Bing Zhang [1], Lu Wang [3,5✉] & Lin Yuan [1✉]

The quality and application of super-resolution fluorescence imaging greatly lie in the dyes' properties, including photostability, brightness, and Stokes shift. Here we report a synergistic strategy to simultaneously improve such properties of regular fluorophores. Introduction of quinoxaline motif with fine-tuned electron density to conventional rhodamines generates new dyes with vibration structure and inhibited twisted-intramolecular-charge-transfer (TICT) formation synchronously, thus increasing the brightness and photostability while enlarging Stokes shift. The new fluorophore **YL578** exhibits around twofold greater brightness and Stokes shift than its parental fluorophore, Rhodamine B. Importantly, in Stimulated Emission Depletion (STED) microscopy, **YL578** derived probe possesses a superior photostability and thus renders threefold more frames than carbopyronine based probes (CPY-Halo and 580CP-Halo), known as photostable fluorophores for STED imaging. Furthermore, the strategy is well generalized to offer a new class of bright and photostable fluorescent probes with long Stokes shift (up to 136 nm) for bioimaging and biosensing.

[1] State Key Laboratory of Chemo/Biosensing and Chemometrics, College of Chemistry and Chemical Engineering, Hunan University, Changsha 410082, China. [2] Optical Microscopy Facility, Max Planck Institute for Medical Research, Heidelberg 69120, Germany. [3] Department of Chemical Biology, Max Planck Institute for Medical Research, Heidelberg 69120, Germany. [4] Institute of Chemical Sciences and Engineering, École Polytechnique Fédérale de Lausanne (EPFL), Lausanne CH-1015, Switzerland. [5] Key Laboratory of Smart Drug Delivery, Ministry of Education, School of Pharmacy, Fudan University Shanghai 201203, China. [6]These authors contributed equally: Gangwei Jiang, Tian-Bing Ren. ✉email: lwangfd@fudan.edu.cn; lyuan@hnu.edu.cn

Super-resolution microscopy that breaks the diffraction limit of light offers a powerful tool to investigate cellular biology on a molecular scale[1–3]. The combination of Stimulated Emission Depletion (STED) microscopy and suitable labeling strategies enables the real-time observation of cellular structures and biological processes with high spatial and temporal resolution[4,5]. However, the high-powered depletion laser beam in STED imaging causes severe photobleaching, resulting in more stringent requirements for fluorescent probes[6,7]. The ideal fluorescent probe for super-resolution imaging should possess (1) high brightness to enable visualization of low-abundant targets, (2) good photostability to permit long-term tracking and high-fidelity imaging under high-powered STED laser, (3) long Stokes shift to increase signal-to-noise ratio and reduce re-excitation caused by the STED light[8,9], (4) various excitation/emission wavelength with short and long Stokes shift, to enable monitoring of multi-targets at the same time[3,8,10]. In addition, cell permeable dyes are also highly desired in live-cell imaging to avoid the potential interference and artificial errors from the sample fixation or invasive techniques, such as bead loading[11] and microinjection[12]. However, the development of such probes is particularly challenging and thus the ideal probe has not yet been reported.

Rhodamine and its derivatives are the most popular fluorophores applied in STED microscopy due to their exceptional properties, including high brightness, good photostability, and various excitation/emission wavelength[3]. Recently, some practical strategies have been reported to further improve their properties, including brightness[13–18], cell-permeability[19,20], fluorogenicity[19–22], photostability[16,18,23], and Stokes Shift[9,24–26], to meet the requirements of various fluorescence imaging techniques. In the excited state of regular rhodamines (e.g., rhodamine B (RhB)) the dialkylamine groups form the twisted-internal-charge-transfer (TICT) state through a C-N bond rotation, which leads to energy relaxation without emission of a photon but with a rapid nonradiative decay (Fig. 1)[27]. The nonradiative state can also produce radicals after reaction with molecular oxygen and often cause photobluing, photobleaching, and fluorescence signal blinking[6,23,27,28] (Supplementary Fig. 1). Thus, inhibition of the TICT effect by reducing or preventing the rotation of the C-N

bond in rhodamines (e.g., replacing dialkylamino with 7-azabicyclo[2.2.1]heptane[18], azetidine, or aziridine[16]) was utilized to improve brightness and photostability (Fig. 1). Another method is to increase the energy barrier of forming TICT state by reducing electron density in fluorophore scaffold by introducing electron-withdrawing groups (EWGs) (e.g., quaternary piperazine[15] or sulfone-functionalized piperidine moieties[14]) (Fig. 1). Even such methods have greatly improved the brightness of various fluorophores, the severe photobleaching caused by the high-powered depletion laser in STED microscopy still exists and has become a significant bottleneck[7]. In addition, the above modifications in the symmetric structure of rhodamines usually produce a small Stokes shift (20–30 nm) with a large overlap between the excitation and emission spectra, which often causes self-quenching and low signal-to-noise ratio in fluorescence imaging[29,30]. The smaller Stokes shift also requires more strict filter settings in fluorescence microscopy. To address this problem, some strategies have been proposed to modify rhodamines' scafford. For example, vibronic structure in fluorophores was produced to induce strong vibrational relaxation in the excitation state of fluorophores, thus significantly increasing Stokes shift[25]. However, the brightness of these dyes in an aqueous solution is usually unsatisfying[24,25]. Until now, to the best of our knowledge, no universal strategy was reported to enable the improvement of brightness, photostability, and Stokes shift all at once.

To address the above challenge, we combine the strength of different design methods and propose a synergistic strategy to develop a new type of asymmetric rhodamines (Yue Lu dyes, abbreviated as **YLs**) that combines TICT inhibition and vibronic structure (Fig. 1 and Supplementary Fig. 2). Among these rhodamine derivatives, **YL578** with 2-(2,2,2-trifluoroethyl)octahydropyrrolo[1,2-a]pyrazine moiety shows a significantly enhanced brightness, improved photostability, and expanded Stokes shift while preserving good cell permeability. **YL578**-derived probes show excellent performance in wash-free organelles staining and protein labeling in confocal and STED microscopy. The design strategy is further extended to other widely-used fluorophores, transferring them to a new class of fluorescent probes and biosensors with great brightness, high photostability, and long Stokes shift.

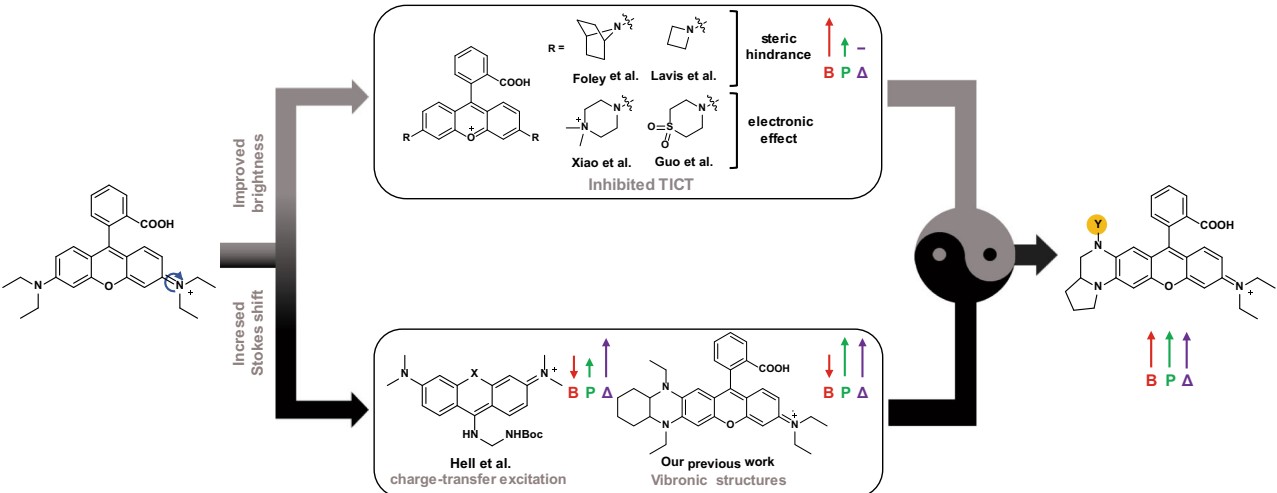

**Fig. 1 The conventional and new strategies to improve the brightness, photostability, and Stokes shift.** In the previous work, brightness, photostability, and Stokes shift can be partially enhanced by inhibiting TICT, generating vibronic structures, or inducing charge-transfer excitation. A new strategy that unites the strength of TICT inhibition and vibronic structure is proposed to simultaneously increase the brightness, photostability, and Stokes shift. Y (orange circle) denotes the positions used to introduce EWGs to tune electron density. B, P, and Δ represent brightness, photostability, and Stokes shift respectively.

## Results

**Rational design of bright and photostable dyes with long Stokes shift.** Recently reported strategies offer two important routes to improve properties of regular rhodamines: symmetric xanthene with inhibition of TICT effect and asymmetric xanthene with vibronic structure (Fig. 1). However, none of the reported approaches can simultaneously improve brightness, photostability, and Stokes shift. We assumed that a combination of vibronic structure and TICT inhibition could open up an opportunity to develop a new type of bright and photostable fluorophores with long Stokes shift (Fig. 1).

We started from the asymmetric rhodamine to enlarge the Stokes shift and **1** was thus synthesized based on the reported work[24,25]. Consistent with previous work[25], **1** displays a greatly increased Stokes shift (99 nm) than RhB (27 nm). However, only a weak fluorescence signal was collected from **1** in PBS buffer solution (Supplementary Fig. 3 and Supplementary Table 1), limiting its application in a cellular environment. Inspired by the reported approaches that optimize the brightness through tuning the electron density in xanthene[14,15,20], we thus developed new dyes **2–7** that contained various EWGs in quinoxaline moiety of the fluorophore scaffold. Such fluorophores can be easily synthesized by the condensation of 2-(4-diethylamino-2-hydroxybenzoyl)benzoic-acid and quinoxaline moieties **S-(1-6)**, which were generated via nucleophilic substitution (**S-(1-5)**) or amidation followed by the reduction reaction (**S-6**) (Fig. 2a and Supplementary Figs. 37, 38).

In the dyes, the emission maxima ($\lambda_{em}$) showed a clear hypochromatic shift from 673 nm to 590 nm as the electron-withdrawing ability of quinoxaline substituents increases, which showed a linear correlation between $\lambda_{em}$ and Hammett substituent constant ($\sigma_p$) (Supplementary Fig. 4). The inductive effect of the substituents was further demonstrated by the well-regulated changing of the electron density of xanthene scaffolds in DFT calculation (Supplementary Fig. 5). Meanwhile, we found that the stronger the electron-withdrawing ability of quinoxaline in **1–6** was, the higher brightness, smaller Stokes shift, and narrower full width at half maximum (FWHM) the fluorophores displayed (Fig. 2b and Supplementary Table 2). Considering the importance of both brightness and Stokes shift, we identified **YL578 (6)** with 2-(2,2,2-trifluoroethyl) octahydropyrrolo[1,2-a] pyrazine group (**S-6**) as the best fluorophore. It displayed a 2.4-fold increase in quantum yield (0.74) and twofold enhancement in brightness ($\varepsilon \times \Phi = 6.6 \times 10^4$ L mol$^{-1}$ cm$^{-1}$) in aqueous solution compared to its parental fluorophore, RhB. Meanwhile, **YL578** showed a red-shifted excitation/emission spectra of 578 nm/634 nm, generating a long Stokes shift of 56 nm (Fig. 2c). In addition, **YL578** displayed significantly higher photostability and less photobluing over RhB and JF549[23] under the illumination of light at 530 nm for 80 min (Fig. 2d and Supplementary Figs. 6–8). We also observed highly consistent fluorescence intensity and lifetime of **YL578** in various solvents or buffer solutions with abundant proteins, which cannot be obtained with RhB (Fig. 2e, f, Supplementary Fig. 9). Importantly, the DFT calculation of **YL578** showed the asymmetric electron distribution in the HOMOs (Fig. 2g) and increased energy barrier to form TICT state (Fig. 2h). To further inhibit TICT formation, we replaced the diethylamino group with azetidine moiety in **YL578**, producing **YL-Az**. Interestingly, **YL-Az** also exhibited red-shift absorption/emission spectra, longer Stokes shift, and improved photostability, which is quite similar to **YL578**. We also synthesized a symmetrical **S-6**-containing dye, **bis-YL**, which shows an increased brightness in EtOH. However, **bis-YL** showed a shorter Stokes shift than **YL578** which indicates the role of asymmetry vibronic structure in enlarging the Stokes shift (Supplementary Fig. 10). These results are consistent with the

DFT calculations, indicating the coincidence of vibronic feature[25] and TICT inhibition[15] in **YL578**. The outstanding performance of **YL578** thus demonstrates the exceptional strength of the proposed synergistic strategy in dye development.

**Evolution of YL578 derived probes for live-cell confocal imaging.** We next incubated HeLa cell with **YL578** to evaluate its cell permeability, brightness, and photostability in live-cell imaging. **YL578** showed a similarly fast cellular staining as RhB, while possessed a stronger fluorescence signal (Supplementary Fig. 11). The significantly increased fluorescence signal is assumed to be due to the combination of higher brightness and larger Stokes shift from **YL578** (Supplementary Fig. 12). Importantly, RhB can be easily photobleached within 10 min under continuous irradiation at 560 nm, while negligible signal change of **YL578** and JF549 was observed, indicating the outstanding photostability of **YL578** (Fig. 3a and Supplementary Fig. 13). To achieve the specific labeling of target protein, we next synthesized **YL578** ligand (**YL578-Halo**) from carboxyl-containing **YL578** (**9**) for labeling HaloTag, a widespread protein tag[31] (Fig. 3b). Incubation of live HeLa cells expressing a HaloTag–histone 2B (H2B) fusion with **YL578-Halo** offered bright nuclear staining (Supplementary Fig. 14), providing 1.5-fold stronger fluorescence signal than RhB derivatives, RhB-Halo (Fig. 3c–f). Interestingly, compared to RhB-Halo, **YL578-Halo** also displayed a much lower cytoplasmic background, which thus endowed a higher nuclei-to-cytosol signal ratio ($F_{nuc}/F_{cyt} = 18$) without any washing steps (Fig. 3c–f), which might result from the transporter-mediated cellular uptake and efflux of different probes[32]. Since it was reported that shifting the equilibrium from the zwitterionic form to the spirocyclic form could help optimize the cell permeability and fluorogenicity[19,20], we next transferred the carboxyl group in **9** to acyl 2,2,2-trifluoroethylamide, producing probe **10** with a pKcyl of 6.71 (Supplementary Fig. 15). $D_{50}$, representing the dielectric constant at which half of the fluorophore population is in the zwitterionic form, has been commonly applied to evaluate the equilibrium between zwitterionic and spirocyclic form[33]. Based on the reported work, **10** with a $D_{50}$ of 45 was thus expected to be a good candidate for generating fluorogenic probes (Supplementary Fig. 16). The subsequent synthesized **10** HaloTag ligand (**10-Halo**) showed a large increase in absorbance (23-fold) and fluorescence intensity (490-fold) upon binding to HaloTag (Supplementary Fig. 17). Consistently, the intracellular background signal of live HeLa cells treated with **10-Halo** is extremely low, thereby offering a superior signal-to-noise ratio in nuclear protein labeling ($F_{nuc}/F_{cyt} = 106$) (Fig. 3e, f). Notably, even the formation of spirolactam in **10** improves the fluorogenicity and cell permeability (Supplementary Fig. 18), it also reduces the brightness to some extent due to the incomplete recovery to the fluorescent zwitterionic state upon HaloTag binding (Fig. 3f)[22].

We next extended this bright and photostable fluorophore **YL578** to prepare new probes for organelles staining in living cells. Probe **YL578-Mito** and **YL578-Lyso** were synthesized by one-step esterification and amidation reaction respectively (Supplementary Figs. 46, 47). We succeeded to utilize these probes in fast and high contrast wash-free staining in mitochondria and lysosomes, as confirmed by the colocalization with MitoTracker Green and LysoTracker Green respectively (Fig. 3g–j and Supplementary Figs. 19, 20).

**Superior photostability of YL578 derivatives in STED imaging.** STED microscopy allows the visualization of biological structures with high spatial and temporal resolution in living cells[34–39]. However, the severe photobleaching of fluorophores has greatly limited the frame number collected in STED imaging. Till now,

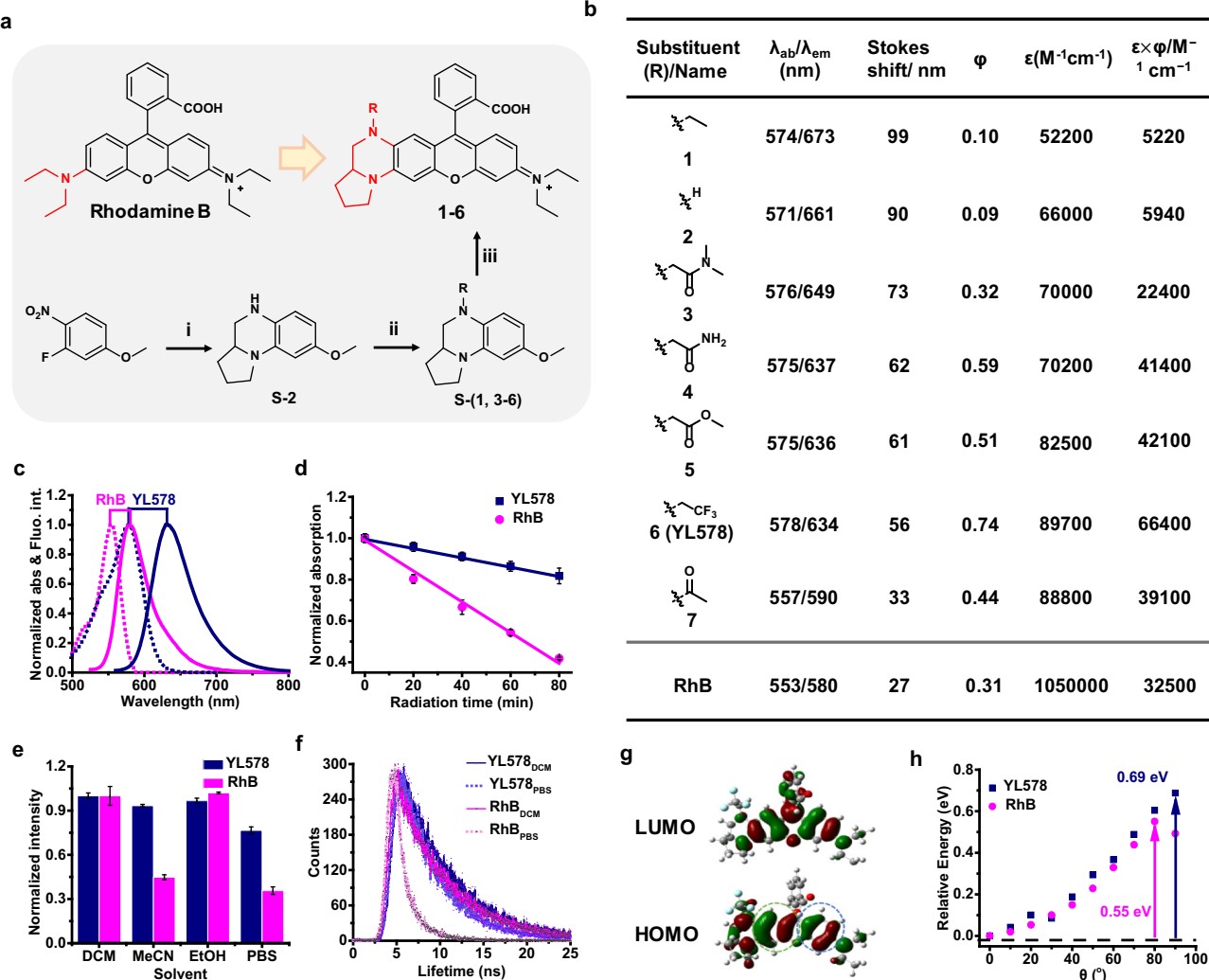

**Fig. 2 Development of YL dyes. a** Synthesis of rhodamine **1–6**. Reaction conditions: i) Proline methyl ester hydrochloride, triethylamine, acetonitrile, reflux, 12 h; methanol, zinc powder, 32% HCl, r.t., 30 min; tetrahydrofuran, NaBH$_4$, boron trifluoride-diethyl etherate, reflux, 1 h; ii) For **2–5**, bromide, K$_2$CO$_3$, acetonitrile, 90 °C, 2 h; for **6 (YL578)**, trifluoroacetic anhydride, tetrahydrofuran, room temperature, 10 min; sodium borohydride, boron trifluoride etherate, reflux, 1 h; iii) 2-(4-Diethylamino-2-hydroxybenzoyl)benzoic acid, methanesulfonic acid, 90 °C, 2 h. The moiety in red indicates the difference between YL dyes and Rhodamine B. **b** The photophysical properties of **1–7** in aqueous solution. λ$_{abs}$, absorption at λ$_{max}$; ε, extinction coefficient; λ$_{em}$, emission at λ$_{max}$; φ, quantum yield; ε × φ, brightness. **c** Normalized absorption and emission spectra of **YL578** and RhB. **d** Absorption at λ$_{max}$ of **YL578** and RhB was plotted as a function of irradiation time with a laser (1 W) at 530 nm. Solution concentrations were adjusted to be comparable to one another in terms of optical density at 530 nm. Error bars, ±s.e.m. $n$ = 3. **e** Normalized fluorescence maxima intensities of **YL578** and RhB in various solvents. Error bars, ±s.e.m. $n$ = 3. **f** Fluorescence lifetime of **YL578** and RhB in DCM and PBS. **g** DFT optimized HOMO and LUMO orbital plots of **YL578**. **h** Calculated potential energy surfaces of **YL578** and RhB in water.

only a very few reported probes for covalently labeling of proteins can provide satisfying performance in STED imaging, especially 3D STED microscopy[40–42]. Rhodamine derivatives (e.g., SiR[33], CPY[37]) are the most popular fluorophores in live-cell STED imaging due to their good photophysical properties and cell permeability[3]. 580CP-Halo[17] and CPY-Halo[17,19], known as photostable STED probes, and JF608-Halo[20], azetidine derivative of CPY-Halo, were selected to compare the performance with **YL578-Halo** in STED microscopy with a 775 nm depletion laser since they share a similar absorption or emission wavelength. **YL578-Halo**, 580CP-Halo, CPY-Halo, and JF608-Halo were incubated to specific stain vimentin fused with HaloTag in U-2 OS cells. 580CP-Halo, CPY-Halo, and JF608-Halo provided STED images with FWHM resolution of 116 ± 6, 86 ± 9, and 83 ± 10 nm in the first frame. However, only 2–3 frames of STED images with >50% of the initial fluorescence intensity were

obtained due to the rapid photobleaching under a 775 nm depletion laser (Fig. 4a, b and Supplementary Figs. 21–23). In contrast, under the identical conditions, **YL578-Halo** offers 9 frames of STED images with >50% of the initial fluorescence intensity while remaining FWHM resolution of 57 ± 5 nm (Fig. 4a, b and Supplementary Fig. 24). When optimizing the imaging settings to obtain the highest resolution, **YL578-Halo** enabled the visualization of vimentin filaments with an FWHM of 37 ± 4 nm (Fig. 4c and Supplementary Fig. 25). The significantly increased frame numbers and resolution in STED imaging demonstrate the superior photostability of **YL578-Halo**. We thus tried to utilize **YL578-Halo** in 3D STED imaging, which is difficult to be achieved when rapid photobleaching of traditional fluorophores occurs during sequential *xzy*-scan[39]. Incubation of U-2 OS cells transiently expressing mitochondrial import receptor Tomm20-HaloTag with **YL578-Halo** enabled the

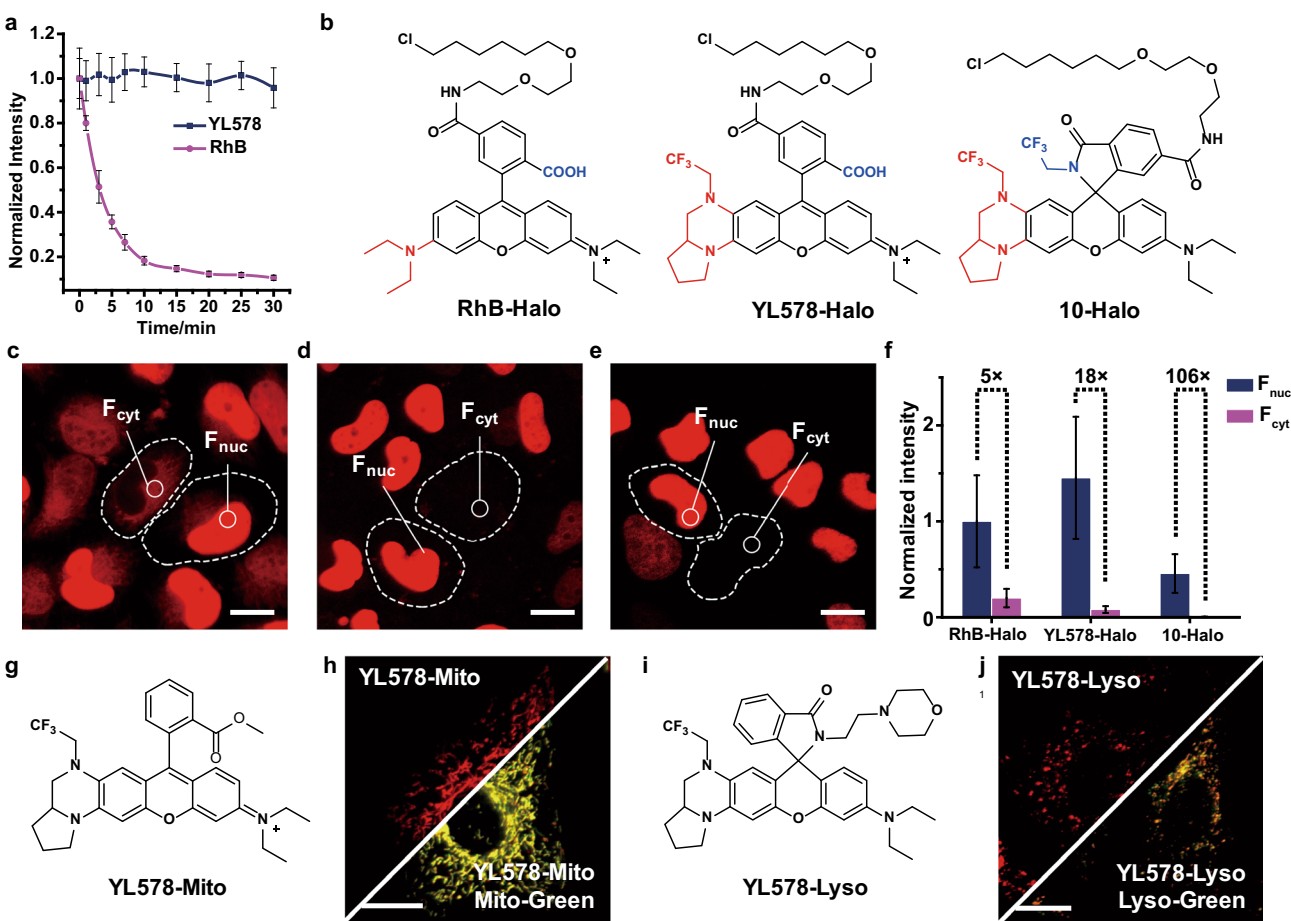

**Fig. 3 Utility of YL derivatives in live-cell imaging. a** Comparison of the photostability of 5 μM **YL578** or RhB in live HeLa cells with continuous irradiation at 560 nm in confocal microscopy. Error bars, ±s.e.m. 50 cells were examined in three independent experiments separately. **b** Structures of HaloTag ligands RhB-Halo, **YL578-Halo**, and **10-Halo**. The moiety in red and blue indicate the difference between the three probes. Live-cell, no-wash confocal images (60×) of co-cultured H2B-HaloTag-expressing HeLa cells and wild-type HeLa cells with 250 nM RhB-Halo (**c**), **YL578-Halo** (**d**), and **10-Halo** (**e**), the dashed lines represent the cellular boundary. **f** Fluorescence ratio ($F_{nuc}/F_{cyt}$) of RhB-Halo, **YL578-Halo**, and **10-Halo** in live-cell confocal microscopy. Bar plot representing the normalized nuclear signal ($F_{nuc}$, H2B-Halo-expressed HeLa cells) and the cytosolic signal ($F_{cyt}$, wild-type HeLa cells). Fluorescence intensities were normalized to the nuclear signal of RhB-Halo. Error bars, ±s.e.m. 50 cells were examined in three independent experiments separately. **g** Structure of **YL578-Mito**. **h** Live-cell, no-wash confocal images of co-incubated HeLa cells with **YL578-Mito** and Mito-Green. **i** Structure of **YL578-Lyso**. **j** Live-cell, no-wash confocal images of co-incubated HeLa cells with **YL578-Lyso** and Lyso-Green. Scale bar, 20 μm.

construction of 3D STED images of Tomm20 along the whole mitochondria (Fig. 4d and Supplementary Fig. 26). In addition, the excellent cell permeability and high contrast staining utilizing **YL578-Halo** and **10-Halo** allow us to perform the wash-free live-cell STED imaging of vimentin filaments with a resolution of $59 ± 7$ nm (Supplementary Fig. 27). To further prove the benefits of the long Stokes shift probe for super-resolution, we performed multi-color imaging of vimentin filaments, microtubes, F-actins, or DNA through combining **YL578-Halo** with traditional short-Stokes-shift probes (MaP555-actin[19], SiR-DNA[43], and GeR-tubulin[44]) with a single 775 nm STED light (Supplementary Fig. 28).

**Extension of the strategy to different types of fluorophores.** Numerous classic fluorophore scaffolds contain the dialkylamino motif and often suffer from TICT, leading to decreased quantum efficiency and photostability[15,16]. Encouraged by the excellent performance of **YL578**, we next extended the strategy to other widely-used fluorophores with different colors. Replacing the dialkylamine with 2-(2,2,2-trifluoroethyl)octahydropyrrolo[1,2-a] pyrazine in the xanthene of rhodol (**11**) vastly increased the quantum yield from 0.21 to 0.62 and the brightness from

$1.3 × 10^4$ to $3.2 × 10^4$ L mol$^{-1}$ cm$^{-1}$. Meanwhile, **11** displayed a red-shifted absorbance/emission maxima from 518 nm/546 nm to 548 nm/612 nm with a longStokes shift of 64 nm (Table 1, Supplementary Table 1, and Supplementary Fig. 29). The confocal imaging of live HeLa cells also indicated the greatly improved photostability of **11** than its parental fluorophore (Supplementary Fig. 30). Importantly, similar improvements in the brightness, photostability, and Stokes shift were also found in other xanthene-containing fluorophores, such as pyronin (Table 1 and Supplementary Figs. 29, 31). We next applied the strategy to coumarin and Boranil which possess very different fluorophore scaffolds. Consistently, the introduction of 2-(2,2,2-trifluoroethyl) octahydropyrrolo[1,2-a]pyrazine motif in **13–15** also offers dramatically increased brightness (5.1–8.1 folds) and improved photostability (Supplementary Figs. 29, 32, 33 and Supplementary Table 1). Notably, **14** and **15** displayed a large Stokes shift of 92 nm and 136 nm, respectively (Table 1). Chloroalkane was next conjugated with **16**, an analog of **14**, to produce HaloTag ligand **16-Halo** (Fig. 5a and Supplementary Fig. 1). Similar to the optical properties of **14**, **16-Halo** also exhibited a large Stokes shift (110 nm) in an aqueous solution (Fig. 5b). Interestingly, probe **16-Halo** displayed a good nuclear protein labeling without any

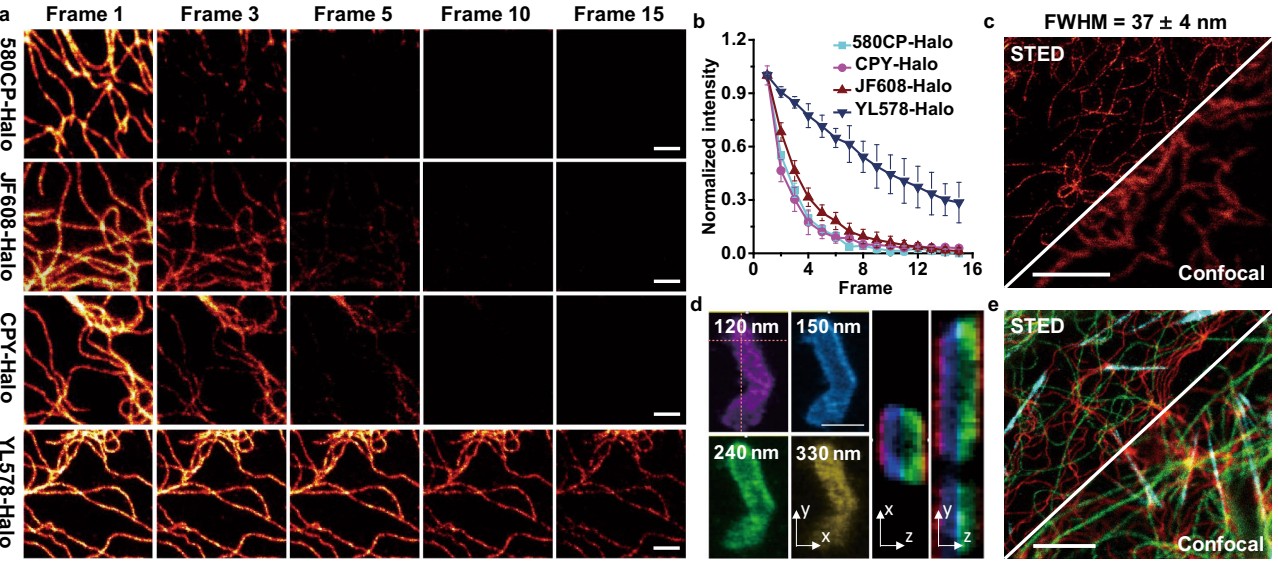

**Fig. 4 Super-photostable YL578-Halo in 3D and live-cell STED microscopy. a** Multiframe STED imaging of fixed U-2 OS vimentin-HaloTag-expressing cells labeled with 50 nM CPY-Halo, JF608-Halo, 580CP-Halo and **YL578-Halo** respectively (STED at 775 nm). Scale bar, 1 μm. **b** Normalized fluorescence intensities of vimentin filaments labeled with CPY-Halo, JF608-Halo, 580CP-Halo, and **YL578-Halo** plotted as a function of frame numbers in STED images. Error bars, ±s.e.m. 50 cells were examined in three independent experiments separately. **c** Confocal and STED images of live U-2 OS stably expressing vimentin-HaloTag-expressing cells labeled with 50 nM **YL578-Halo**. Scale bar, 5 μm. **d** 3D STED images of mitochondria in U-2 OS cells expressing Tomm20-Halo labeled with 50 nM **YL578-Halo**. The images were recorded in sequential *xzy*-scanning mode. STED images recorded along the *z*-axis with a step of 35 nm were shown in different colors. Dashed lines indicate the position of XZ and YZ cross sections shown in the right image panel. Scale bar, 1 μm. **e** Three-color confocal and STED images of live U-2 OS Vimentin-HaloTag-expressing cells labeled with 500 nM **YL578-Halo** (red, STED at 775 nm), 500 nM MaP555-actin (cyan, STED at 775 nm) and 500 nM GeR-tubulin (green, STED at 775 nm). Scale bar, 5 μm.

washing steps, while its parental probe Coumarin-Halo showed no specific staining under the same conditions (Fig. 5c, d). Furthermore, **16-Halo** enables the imaging of vimentin filaments in live-cell STED microscopy using a depletion laser of 595 nm (Supplementary Fig. 34). Overall, these results demonstrate that replacing the dialkylamino motif with 2-(2,2,2-trifluoroethyl) octahydropyrrolo[1,2-a]pyrazine is generalizable to different fluorophore scaffolds, producing substantial improvements in brightness, photostability, and Stokes shift simultaneously.

Bright and photostable fluorophores are always highly desired to develop chemical sensors to avoid the false signal from fluorescence photobleaching[45,46]. It is worth noting that the new strategy only modifies the dialkylamino motif on one side of the xanthene scaffold in Rhodol, which leaves the oxygen atom on the other side free for producing sensors[47,48]. In proof-of-principle experiments, the phosphate group was conjugated to **11** to develop a sensor for qualitatively detecting alkaline phosphatase (ALP) (Fig. 5e). **11-ALP** showed ultra-weak fluorescence in buffer solution, while the addition of alkaline phosphatase can remove the phosphate group, thereby resulting in a huge signal enhancement (Fig. 5f, g). **11-ALP** also displayed good specificity towards ALP in the presence of various biomolecules (Fig. 5h). HeLa cells incubated with **11-ALP** showed a much stronger fluorescence signal than normal liver L02 cell line (Fig. 5i and Supplementary Fig. 35), probably due to the higher expression level of ALP in tumor cells[49]. Meanwhile, the addition of $Na_3VO_4$ reduced the ALP level in HeLa cells, thus producing a decreased fluorescence signal (Fig. 5i and Supplementary Fig. 35).

## Discussion
The chemical structures of fluorophores determine their properties (e.g., brightness, photostability, Stokes shift, wavelength, and cell permeability). However, the reported modification methods can only partially improve such key properties[3]. Meanwhile, new imaging techniques have a higher requirement

for fluorescent probes, such as ultra-high photostability for STED microscopy[6,50].

To improve such properties simultaneously, we proposed a new fluorophore scaffold modification method *via* the combination of vibronic structure and TICT inhibition. After systematic modification and testing (Fig. 2), 2-(2,2,2-trifluoroethyl) octahydropyrrolo[1,2-a]pyrazine motif was successfully developed and identified as a unique group to simultaneously improve brightness, photostability, and Stokes shift. The vibronic feature and TICT inhibition from 2-(2,2,2-trifluoroethyl) octahydropyrrolo[1,2-a]pyrazine motif were also confirmed by DFT calculation (Fig. 2g, h), supporting the assumption of the synergistic strategy. We thus believe that the increased Stokes shift is mainly from the vibronic structure, while the enhanced brightness results from TICT inhibition and the decreasing electron density of quinoxaline moiety, indicated by the significantly higher quantum yields of **11–16** than **17–21** (Supplementary Table 1). The 2-(2,2,2-trifluoroethyl) octahydropyrrolo[1,2-a]pyrazine motif in **YL578** is assumed to intensify vibration relaxation in the excitation state while decrease the non-radiative decay between excitation state and ground state, thus causes higher brightness with longer Stokes shift. Interestingly, **YL-Az** also displayed red-shifted absorption/emission spectra, larger Stokes shift than JF549, but cannot provide largely improved brightness than **YL578** (Supplementary Fig. 13). Notably, **YL578** exhibits surprisingly high photostability in STED microscopy, which even permits 3-time more frame numbers than 580CP, CPY, and JF608. TICT state is typically nonemissive and highly reactive[51], while introduction of 2-(2,2,2-trifluoroethyl)octahydropyrrolo[1,2-a]pyrazine group can efficiently inhibit TICT formation (Fig. 1h), thus reducing photobleaching. Meanwhile, it is worth noting that the enhanced photostability was also found in rhodamines with vibronic structure, indicating that the superior photostability of **YL578** could be the synergetic effect of TICT inhibition and vibronic structure[15,16]. The success of simultaneously improving several properties of classic fluorophores in this

**Table 1 Spectroscopic data of 11–15 and reference fluorophores R-(2–6)[56,57].**

| Dye structure | Name | $\lambda_{ab}$(nm) | $\lambda_{em}$(nm) | Stokes shift/nm | $\varphi$ | $\varepsilon$(M$^{-1}$cm$^{-1}$) | $\varepsilon \times \varphi$/M$^{-1}$cm$^{-1}$ | $\Delta(\varepsilon \times \varphi)$ |
|---|---|---|---|---|---|---|---|---|
|  **11** / **R-2** | 11 | 548[a] | 612[a] | 64 | 0.62[a] | 51,000[a] | 31,620 | 2.5 |
| | R-2 | 518[56] | 546[56] | 29 | 0.21[56] | 60,000[56] | 12,600 | 1 |
|  **12** / **R-3** | 12 | 577[a] | 623[a] | 46 | 0.71[a] | 83,300[a] | 59,140 | 3.2 |
| | R-3 | 552[57] | 581[57] | 24 | 0.18[57] | 103,000[57] | 18,540 | 1 |
|  **13** / **R-4** | 13 | 406[b] | 513[b] | 117 | 0.55[b] | 32,400[b] | 17,820 | 5.2 |
| | R-4 | 372[b] | 470[b] | 98 | 0.19[b] | 18,000[b] | 3420 | 1 |
|  **14** / **R-5** | 14 | 464[b] | 556[b] | 92 | 0.67[b] | 26,200[b] | 17,550 | 8.1 |
| | R-5 | 430[b] | 484[b] | 54 | 0.06[b] | 32,200[b] | 2170 | 1 |
|  **15** / **R-6** | 15 | 439[b] | 575[b] | 136 | 0.40[b] | 25,100[b] | 10,000 | 8.0 |
| | R-6 | 401[b] | 462[b] | 61 | 0.04[b] | 31,300[b] | 1250 | 1 |

[a]Measured in PBS (25 mM, pH 7.4).
[b]Measured in PBS (25 mM, pH 7.4) containing 20% EtOH. $\Delta(\varepsilon \times \varphi)$ represents the ratio of the increased brightness.

work demonstrated the extraordinary strength of the synergistic strategy, which will boost many new design strategies to create the next generation of fluorophores.

RhB is a classic fluorophore that has been utilized to develop numerous fluorescent probes and biosensors[52,53]. We first applied this synergistic strategy to transfer RhB to new dye **YL578**, which exhibits around twofold greater in brightness ($\varepsilon \times \Phi = 6.6 \times 10^4$ L mol$^{-1}$ cm$^{-1}$) and Stokes shift (56 nm) (Fig. 2). Importantly, **YL578** displayed exceptional photostability, thus allowing us to perform 3D super-resolution imaging, which is very difficult to be achieved with regular fluorophores (Fig. 3). Notably, **YL578** also showed excellent performance in two-photon microscopy (Supplementary Fig. 36). Since the dialkylamino group is found in most classic fluorophores, this new design strategy can be easily extended to other regular dyes. It is worth noting that the significant improvements were obtained not only in fluorophores containing xanthene scaffold (rhodol, pyronin) but also in the ones with quite distinct scaffolds (coumarin and Boranil) (Table 1). Interestingly, this design strategy is even more effective for coumarin- and Boranil-derived fluorophore **14** and **15**, which showed around 8-fold increased brightness and dramatically enlarged Stokes shifts of (96–136 nm) in aqueous solution (Table 1).

This design strategy utilizes 2-(2,2,2-trifluoroethyl)octahydropyrrolo[1,2-a]pyrazine group to replace the dialkylamino group in fluorophores. Notably, this strategy only requires structure modification on one side of the xanthene scaffold, thus leaving the other amino or hydroxyl group on the other side free to be functionalized. As a proof of concept, **11** was successfully developed for in vitro and in cellulo ALP sensing. The generality and versatility of the design strategy pave the way to improve many other already existing fluorescent probes and create numerous new ones.

## Methods

Detailed synthesis procedures, spectroscopic data of all compounds are provided in the Supplementary Information.

**Materials and general methods**. Unless otherwise stated, all chemical and biological reagents were purchased from commercial suppliers in analytical grade and used without further purification. All tests are performed at room temperature. $^1$H and $^{13}$C nuclear magnetic resonance (NMR) spectra were recorded on a Bruker DRX-400 spectrometer. Mass spectra were recorded on a Matrix-Assisted Laser Desorption Ionization Time of Flight Mass Spectrometry (ultrafleXtreme) or LCQ Advantage ion trap mass spectrometer from Thermo Finnigan or Agilent 1100 HPLC/MS.E.M. spectrometer. UV absorption and emission spectra were recorded on UV-1800 spectrophotometer (Shimadzu Corporation, Japan) and Hitachi F-4600 spectrofluorometer (Tokyo, Japan) respectively. Confocal microscopy was performed on a Nikon A1 plus confocal microscope. STED microscopy was performed on an Abberior Instruments STED 775/595/RESOLFT Expert Line microscope or an Abberior Instruments STED 775/660/multiphoton Infinity Line microscope. CPY-Halo and JF608-Halo are synthesized based on Hell's work[17,44] and Lavis's work[20].

**Confocal live-cell imaging**. HeLa cells were incubated with the DMEM containing streptomycin (100 μg/mL), penicillin (100 U/mL), and 10% FBS at 37°C under 5 wt%/vol CO$_2$ for 24 h. 5 μL stock solution (1 mM) of **1–21** was added to 1 mL culture medium (final probe concentration: 5 μM), with which the live HeLa cells was incubated for 30 min before cell imaging. To evaluate ALP levels in different cell lines, live HeLa and L02 cells were cultured in 1 mL phenol-red-free DMEM containing 5 μM **11-ALP** for 30 min before imaging. In the control

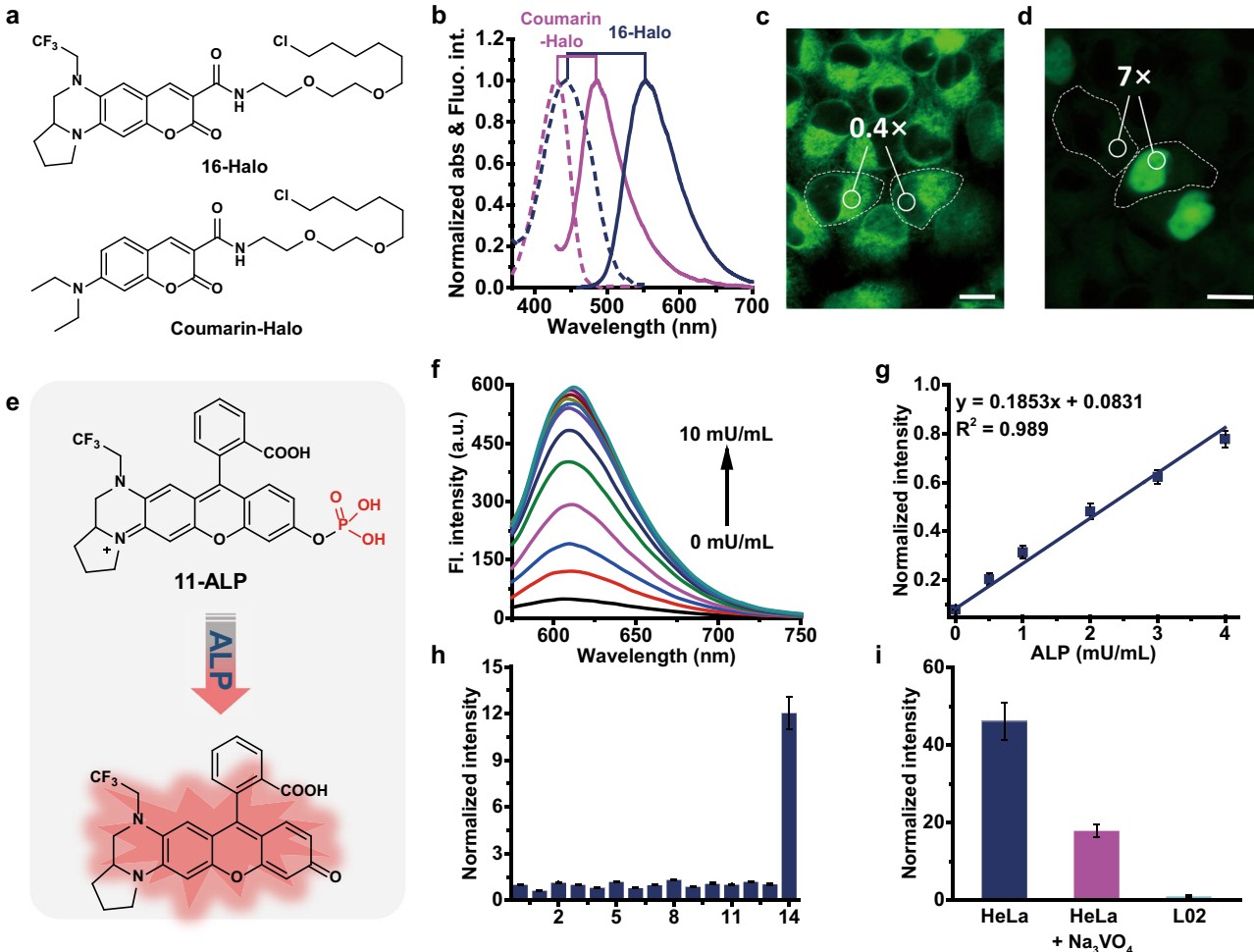

**Fig. 5 Application of YL derivatives in bioimaging and biosensing. a** Structures of **16-Halo** and Coumarin-Halo. **b** Normalized absorption and emission spectra of **16-Halo** and Coumarin-Halo. No-wash live-cell confocal images of co-cultured normal HeLa cells and HeLa cells expressing H2B-Halo labeled with 250 nM Coumarin-Halo (**c**) or **16-Halo** (**d**). Scar bar, 20 μm. **e** Structure of **11-ALP** and its sensing mechanism. The moiety in red in **11-ALP** indicates the response site of ALP. **f** The fluorescence spectra of **11-ALP** upon addition of ALP (0–10 mU/mL) in 10 mM tris-HCl buffer (pH 7.4). $\lambda_{ex}$ = 560 nm. **g** Linear fitting curve of **11-ALP** at 615 nm against ALP from 0 to 4 mU/mL. Error bars, ±s.e.m. $n$ = 3. **h** Normalized fluorescence responses of **11-ALP** (5 μM) to various biomolecules: (0) probe alone (5 μM); (1) Na$^+$ (100 μM); (2) K$^+$ (100 μM); (3) Cysteine (100 μM); (4) Homocysteine (100 μM); (5) GSH (1 mM); (6) H$_2$S (100 μM); (7) H$_2$O$_2$ (100 μM); (8) ONOO$^-$ (5 μM); (9) HClO (5 μM); (10) Butyrylcholinesterase (BchE, 10 mU/mL); (11) Acetylcholinesterase (AchE, 10 mU/mL); (12) Esterase (10 mU/mL); (13) Nitroreductase (NTR, 400 mU/mL); (14) Alkaline phosphatase (ALP, 5 mU/mL). Error bars, ±s.e.m. $n$ = 3. **i** Evaluation of **11-ALP** in sensing ALP of live L02 cells and HeLa cells. Live L02 cells and HeLa Cells were incubated with **11-ALP** for 30 min, or pre-incubated with 200 μM Na$_3$VO$_4$ for 1 h prior to imaging. Error bars, ±s.e.m. $n$ = 10.

experiment, HeLa cells were firstly cultured with Na$_3$VO$_4$ (1 mM) for 1 h, then treated with **11-ALP** (5 μL 1 mM stock solutions) and incubated for 30 min before imaging.

To express H2B-Halo in living cells, plasmid LZ10 pbrebac-H2B-Halo (#91564, addgene) was transfected into HeLa cells utilizing lipo8000 (Beyotime Biotechnology) following the standard protocols[54]. Live HeLa cells were incubated with 250 nM probe for 0.5–5 h at 37 °C in a 5% CO$_2$ atmosphere and directly imaged using confocal microscopy without washing steps unless specifically stated. Microscopy conditions: RhB, RhB-Halo, R-3 and **YL578, 11, 12, YL578-Halo, 10-Halo, 11-ALP**, $\lambda_{ex}$ 561 nm; detection range 585–675 nm. R-2 and **14, 16-Halo**, $\lambda_{ex}$ 488 nm; detection range 500–550 nm. **13**, $\lambda_{ex}$ 405 nm; detection range 500–550 nm. Coumarin-Halo and R 4–6, $\lambda_{ex}$ 405 nm; detection range 425–475 nm. **15**, $\lambda_{ex}$ 405 nm; detection range 570–620 nm.

**The photostability testing in STED microscopy**. U-2 OS stably expressing Vimentin-HaloTag cells[55] were seeded on glass coverslips and incubated in imaging medium that contained 50 nM **YL578-Halo**, 580CP-Halo, CPY-Halo, and JF608 for 6 h at 37 °C. Next, the cells were fixed with 4% PFA for 20 min and quenched for 5 min in NH$_4$Cl and glycine (both 100 mM), washed in PBS, and mounted in Mowiol. The fixed samples were imaged on an Infinity Line STED microscope equipped with 775 and 660 STED lines, and 518 nm, 561 nm, 640 nm, and multiphoton excitation lines (Abberior Instruments GmbH). Imaging conditions: $\lambda_{ex}$ 561 nm, detection range 570–700 nm, STED laser 775 nm.

**3D STED microscopy**. Plasmid pcDNA5-FRT-Tomm20-Halo was transfected into the host cell line U-2 OS using Turbofect (ThermoFisher) following the manufacture's protocol. U-2 OS cells transiently expressing Tomm20-HaloTag were next incubated in imaging medium that contained 50 nM **YL578-Halo** overnight at 37 °C. The cells were fixed with 4% PFA for 20 min and then quenched for 5 min in NH$_4$Cl and glycine (100 mM), washed in PBS, and mounted in Mowiol. Imaging was performed on an Expert line STED/RESOLFT system equipped with 595 nm and 775 nm STED lines, and 355 nm, 405 nm, 485 nm, 561 nm, and 775 nm excitation lines (Abberior Instruments). The STED images of mitochondria were collected along the z-axis with a step of 35 nm. The 3D images were constructed in ImageJ with plugin z-stack depth colorcode. Imaging conditions: $\lambda_{ex}$ 561 nm, detection range 570–700 nm, STED laser 775 nm.

**Live-cell STED microscopy**. U-2 OS stably expressing Vimentin-HaloTag cells seeded on glass coverslips were incubated in phenol red-free imaging medium that contained 50 nM **YL578-Halo** or **10-Halo** overnight at 37 °C and then imaged without washing with the 775 nm STED line on the Abberior Instruments Infinity Line microscope. For **16-Halo**, live U-2 OS stably expressing Vimentin-HaloTag cells were treated with 500 nM probe for 90 min and washed with imaging medium. The STED imaging was collected on the Abberior Expert Line system with excitation wavelength at 485 nm and STED lines at 595 nm. Imaging conditions: **YL578-Halo** and **10-Halo**, $\lambda_{ex}$ 561 nm, detection range 570–700 nm, STED laser 775 nm. **16-Halo**, $\lambda_{ex}$ 485 nm, detection range 505–550 nm, STED laser 595 nm.

**Reporting summary**. Further information on research design is available in the Nature Research Reporting Summary linked to this article.

## Data availability

The source data underlying Figs. 2d–g, 3a, f, 4b, 5g–i and Supplementary Figs. 9, 11, 13d, 18b, 21–25, 27 are recorded in a Source Data file. The authors declare that other data related to this research are available within the paper and its Supplementary Information, or from the authors upon reasonable request.

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

## Acknowledgements

This work was supported by the National Natural Science Foundation of China (Project: 21877029 and 22074036 to L.Y., 22004033 to T.R., 32171360 and 22107020 to L.W.) and start-up fund from Fudan University (to L.W.). We thank Jasmine Hubrich for the support with cell culture and Dr. Vladimir Belov and Grazvydas Lukinavicious for providing probe 580CP-Halo and GeR-tubulin.

## Author contributions

L.Y. lead the project. L.Y., L.W., and T.R. conceived and designed the project. L.Y., T.R., K.J., and X.Z. supervised and supported the project. G.J. performed the probe syntheses and spectral experiments. G.J., M.X., and B.X. performed confocal imaging experiments. E.D. conducted the STED imaging experiment. The manuscript was written by G.J., L.W., L.Y., and T.R. and edited by all the coauthors. All authors discussed the results and commented on the manuscript.

## Competing interests

L.Y., T.R., G.J., and X.Z. are inventors of the patent (Patent Application 2021111706051, pending), which was filed by Hunan university. The other authors declare no competing interests.
