## [Peer Review File · Nature Communications]

A synergistic strategy to develop photostable and bright dyes with long Stokes shift for nanoscopyREVIEWER COMMENTS

Reviewer #1 (Remarks to the Author):

In this manuscript, authors reported a synergistic strategy to simultaneously improve the fluorescence brightness, photostability, and Stokes shift of a broad range of regular fluorophores, especially rhodamines. The key of the strategy is incorporating an electron-withdrawing group-functionalized quinoxaline motif, i.e. 2-(2,2,2-trifluoroethyl) octahydropyrrolo[1,2-a]pyrazine group, into regular fluorophore backbone, which not only causes large Stokes shifts due to creating asymmetric electronic structure, but also improves brightness due to inhibiting the formation of TICT state. The resulting fluorophore BDQF-6 exhibits ca. 2-fold greater brightness and Stokes shift (56 nm) than its parental Rhodamine B fluorophore. Based on the dye platform, authors developed HaloTag ligands, and demonstrated the superior signal-to-noise ratios in nuclear protein labeling. Impressively, in STED microscopy, due to the high photostability, BDQF-6 derived probe BDQF-9-Halo renders 3-fold more frames than carbopyronine- and JF608-based probes that are known as photostable fluorophores for STED imaging. Authors also confirmed that the strategy is generalizable to several other types of fluorophores (pyronin, rhodol, coumarin, and Boranil), offering a new class of bright and photostable fluorescent probes with large Stokes shift for bioimaging and biosensing. Overall, the work is attractive and represents an obvious improvement in fluorescence dye development. Thus, I recommend it to be published in the journal after minor revision as follows.

(1) Line 163: "Fcyt/Fnuc = 18" should be corrected to "Fnuc/Fcyt = 18".

(2) The equilibrium between zwitterionic and spirocyclic form should also be evaluated in varied pH values to achieve PK_{cyt} values of BDQF-6 or its derivatives, in order to probe their possible applications in other single-molecule localization microscopy techniques such as dSTORM.

(3) Compared to the improvement in quantum yield and Stokes shift, the improvement in photostability for BDQF-6 is somewhat hard to understand. Is it due to the inhibition of TICT? Authors should give rise to a more detailed mechanism illustration.

Reviewer #2 (Remarks to the Author):

In this work, Jiang and coworkers mainly reported a new group of Rhodamine fluorescent dyes with improved brightness, photostability and Stokes Shift, which are revealed to be suitable for STED imaging in live cells. I think it is very interesting for the researchers in STED super resolution imaging, because a new STED fluorescent dye with excellent photo properties for live cell imaging and this kind of modifications on Rhodamine dye have not yet been reported before. Therefore, I think this work is novel and can be considered to be published in Nature communications after addressing some concerns as follows:

(1) In the introduction section, the authors pointed out that an ideal fluorescent dye should satisfy some standards for STED imaging, high brightness, good photostability and different excitation and emission wavelength can be acceptable, the large Stokes Shift is also a beneficial point for STED, why?

(2) In Figure 1, the authors depicted a scheme that the modification from Rhodamine B to the target dyes, all parameters of the brightness, photostability and Stokes Shift are increased, which seems to be a contradiction. The fluorescent brightness and Photostability are increased, due to the introduction of the rigid scaffold into the molecule to reduce the energy loss, which are easy to be understood. However, the increase of the Stokes Shift seems to be contradictory with the reduction of the energy loss, the authors should make more detailed discussion here.

Reviewer #3 (Remarks to the Author):

In their manuscript titled "A synergistic strategy to develop photostable and bright dyes with long Stokes shift for super-resolution microscopy", Jiang, Ren, and coworkers describe a strategy to increase the brightness, photostability, and Stokes shift of rhodamines and other fluorophore scaffolds through the incorporation of a bicyclic quinoxaline moiety in place of the standard N,N-dialkylamino group common to many fluorescent small molecules. The authors previously reported a series of quinoxaline fluorophores (rhodols in Chen et al. *ACIE* 2017, 56, 16611; rhodamines, pyronines, and oxazines in Ren et al. *JACS* 2018, 140, 7716) that displayed large Stokes shifts but poor quantum yields. The main structural modification here is the fusion of the second 3'-nitrogen substituent onto the quinoxaline to form an octahydropyrrolo[1,2-a]pyrazine, which—like the previous quinoxalines—increases the Stokes shift through the generation of vibronic structures. Although this ring system was reported in the 2017 *ACIE* paper, the authors show here that substituting the 2'-nitrogen with electron-withdrawing groups can rescue quantum yield (presumably by reducing TICT). The substituent best balancing brightness and Stokes shift (since EWGs do decrease the shift) is 2,2,2-trifluoroethyl. The resulting rhodamine "BDQF-6" shows improved QY and photostability compared to rhodamine B, surprising contrast as a self-labeling tag (HaloTag) ligand, and good behavior as a STED dye compared to two other rhodamines with similar spectra/wavelengths (see (3) below). Incorporation of the same ring system into rhodol, pyronine, coumarin, and Boranil dyes also significantly increases brightness and Stokes shift compared to the parent (N,N-dialkylamino) fluorophores.

The main advancement touted in this manuscript is interesting and potentially useful to the chemistry and microscopy communities. However, there are several key issues that should be addressed before this article can be considered for publication in *Nature Communications*:

-Major issues-

(1) The authors emphasize that the key structural modification (the octahydropyrrolo[1,2-a]pyrazine with electron-withdrawing groups) unites two strategies to improve fluorophores: vibronic structures and TICT inhibition. And it does that, to an extent. Why, though, is the "other" side of the rhodamines ignored? Why do all the rhodamines retain the poor, suboptimal N,N-diethyl group on the opposite side of the rhodamine? This moiety is a known liability from both a QY (TICT) and photostability (dealkylation) perspective. The authors should incorporate one or more of the known strategies to lessen TICT and improve brightness—several of which they themselves show in the abstract—such as the work of Foley (azabicycloheptane) or Lavis (azetidines). Replacing the N,N-diethyl with any number of better choices would retain the asymmetry of the rhodamines, provide better fluorophores for comparison to the state-of-the-art in rhodamine dyes, and demonstrate if all of these strategies are indeed synergistic.

(2) Given work in the past 10-20 years, rhodamine B is far from the state-of-the-art as far as rhodamine brightness and photostability is concerned (and could be considered largely obsolete). There are now far brighter, cell-permeable rhodamines with quantum yields that approach 1. To convincingly demonstrate this strategy as a way to improve brightness and photostability, the authors should compare BDQF-6 to more modern rhodamines such as Hell's tert-butyl rhodamines (Butkevich et al. *JACS* 2019, 141, 981) or Lavis's bis(azetidiny)rhodamines like JF549 (Grimm et al. *NM* 2015, 12, 244).

(3) In demonstrating BDQF-9-Halo as an effective STED dye, the authors compare it to "CPY-Halo" and JF608-Halo. Firstly, the structure of CYP-Halo looks to be incorrect in Supplementary Figure 3; the structure as drawn is actually tetramethylrhodamine-HaloTag ligand (TMR-Halo), but I assume it's supposed to be the carbon analog (tetramethyl-carborrhodamine). The reference cited for CYP-Halo is not correct (it does not describe CYP in any detail). How were CYP-Halo and JF608-Halo synthesized or obtained? It is not described in the methods. Beyond the similarity in emission maximum, it's hard to see why these dyes were chosen. To my knowledge, JF608 has not been characterized as a "photostable dye in STED microscopy". Given the significant difference in absorbance/excitation maxima for BDQF-9 compared to CYP and JF608, should the imaging conditions not be adjusted? Most importantly, how does BDQF-9-Halo compare to the myriad STED rhodamines and carbopyronines described by Hell over the past 10+ years? A more advanced imaging experiment—such as 2-color STED (with a SiR dye, for example)—would also go a long

way to demonstrating the utility of these dyes for complex studies.

(4) Although the Supporting Information is certainly extensive, the synthetic experimentals and chemical characterization data for many of the compounds are not currently of publishable quality. All experimentals should contain masses/volumes for all reagents, as well as a specific mass and yield for the product. Each compound should have a specific yield and "state" (yellow oil, white solid, etc.) rather than just a range of yields for each procedure. The salt forms (free base/inner salt? TFA salt?) and/or counterions (for the pyronine and rhodamine methyl ester) are also not currently specified. Most importantly, though, several of the compounds have NMR spectra that demonstrate inadequate purity (or are otherwise broad and uninformative), with (for the ¹H NMR) suspect integrations that exclude significant impurity peaks and (for the ¹³C NMR) cherry-picking of peaks (e.g., clusters of many smaller peaks with only 1 or 2 picked/reported). Some of the more concerning examples are BDQ-3, BDQF-1, BDQF-5, BDQF-6-Lyso, BDQF-9-Halo, BDQF-10, BDQF-11, and BDQF-11-ALP. What is the reason for the CD₃OD/CDCl₃ solvent mixture used for many of the NMR spectra? Perhaps another solvent would resolve some of the broadness observed in many of the ¹H NMR? For compounds that do not have a ¹³C NMR, another assay of purity (e.g., HPLC trace) should be included.

-Minor issues-

(5) The authors curiously do not report a symmetrical rhodamine with the BDQ-CH₂CF₃ moiety on both sides. Although this dye would likely not display an enhanced Stokes shift due to symmetry, it would be an excellent way to showcase how this strategy specifically addresses TICT compared to existing approaches (i.e., by comparing the "bis-BDQ rhodamine" to Foley's bis(azabicycloheptane)rhodamine, Lavis's JF549, etc.).

(6) It appears that many (if not all) of the enhanced Stokes shift, BDQ-containing dyes have broader spectra than their parent (non-BDQ) analogs. The authors should report FWHM, as the broadened spectra can certainly be a concern for multiplexed imaging.

(7) There does not appear to be a definition of the acronym BDQ/BDQF. What does this represent? If, as in their previous paper, the "DQ" is "decahydroquinoxaline", it should be changed, since these do not contain decahydroquinoxalines. They could be referred to as 1,2,3,4-tetrahydroquinoxalines (includes the rhodamine aryl ring but not the fused 5-membered ring) or octahydropyrrolo[1,2-a]pyrazines.

(8) How do the authors explain the surprisingly high contrast of BDQF-9-Halo (18x)? The no-wash contrast of rhodamine ligands (most notably the SiR-type) is most often attributed to the open-closed equilibrium, where the dye is more closed (lactone, non-fluorescent) in solution but shifts to the open form (zwitterion, fluorescent) upon binding to protein. BDQF-6 (BDQF-9-Halo sans HaloTag ligand), however, has spectral properties consistent with a rhodamine that is quite open ($\epsilon = 88,800$) in solution. Moreover, the dioxane-water titration of BDQF-9-Halo (Supplementary Figure 15c) shows essentially no dependence of fluorescence on dielectric constant (i.e., a high lactone-zwitterion equilibrium constant), and HaloTag binding of BDQF-9-Halo elicits almost no change in fluorescence (Supplementary Figure 16).

(9) There is an overwhelming number of Supplementary Figures and Tables (41 total) in the Supporting Information. Several of them are not referred to at all in the main text. The authors should find a way to condense/organize/pair down these figures to make the SI more navigable/readable. Why are Supplementary Table 1-4 separated into four discreet tables when they all show the same type of information for different dyes?

(10) "Till now" should be "Until now".

POINT-BY-POINT RESPONSE TO THE COMMENTS ON THE MANUSCRIPT “A SYNERGISTIC STRATEGY TO DEVELOP PHOTOSTABLE AND BRIGHT DYES WITH LONG STOKES SHIFT FOR SUPER-RESOLUTION MICROSCOPY”

We are grateful to the reviewers and the editor for their insightful comments and suggestions. In the revised manuscript we have attempted to address all these points and changed the manuscript accordingly. Since the compounds in this work were renamed, we also change the name in the questions to make it understandable. Below we provide a point-by-point response to the comments of the reviewers.

Reviewer comments:

Reviewer #1:

In this manuscript, authors reported a synergistic strategy to simultaneously improve the fluorescence brightness, photostability, and Stokes shift of a broad range of regular fluorophores, especially rhodamines. The key of the strategy is incorporating an electron-withdrawing group-functionalized quinoxaline motif, i.e. 2-(2,2,2-trifluoroethyl) octahydropyrrolo[1,2-a]pyrazine group, into regular fluorophore backbone, which not only causes large Stokes shifts due to creating asymmetric electronic structure, but also improves brightness due to inhibiting the formation of TICT state. The resulting fluorophore BDQF-6 exhibits ca. 2-fold greater brightness and Stokes shift (56 nm) than its parental Rhodamine B fluorophore. Based on the dye platform, authors developed HaloTag ligands, and demonstrated the superior signal-to-noise ratios in nuclear protein labeling. Impressively, in STED microscopy, due to the high photostability, BDQF-6 derived probe BDQF-9-Halo renders 3-fold more frames than carbopyronine- and JF608-based probes that are known as photostable fluorophores for STED imaging. Authors also confirmed that the strategy is generalizable to several other types of fluorophores (pyronin, rhodol, coumarin, and Boranil), offering a new class of bright and photostable fluorescent probes with large Stokes shift for bioimaging and biosensing. Overall, the work is attractive and represents an obvious improvement in fluorescence dye development. Thus, I recommend it to be published in the journal after minor revision as follows.

1. Line 163: " $F_{\text{cyt}}/F_{\text{nuc}} = 18$ " should be corrected to " $F_{\text{nuc}}/F_{\text{cyt}} = 18$ ".

Response: We would like to thank the reviewer for pointing this out. We have corrected the error in the main text as highlighted (page 5, line 4).

2. The equilibrium between zwitterionic and spirocyclic form should also be evaluated in varied pH values to achieve PK_{cyl} values of YL578 (6) or its

derivatives, in order to probe their possible applications in other single-molecule localization microscopy techniques such as dSTORM.

Response: As suggested by the reviewer, we added the absorbance of **6** (YL578), **9**, **10** as a function of pH in the supplementary Fig. 15. It was shown that **6**, **9** mainly maintain a zwitterionic form in pH 4.0 – 12.0. Even **10** displays a pH-dependent absorbance (pK_{cyl} : 6.71), **10-Halo** still shows a relatively high equilibrium of fluorescent zwitterion state upon binding to HaloTag (Fig. 3e). To meet the requirement of dSTORM, electron-rich alkylamine or different tag protein may be needed to further shift the equilibrium to non-fluorescent spirocyclic form. The pH titration was added in the revised manuscript (page 5, line 8-9).

- 3. Compared to the improvement in quantum yield and Stokes shift, the improvement in photostability for YL578 is somewhat hard to understand. Is it due to the inhibition of TICT? Authors should give rise to a more detailed mechanism illustration.*

Response: We are grateful to the reviewer for raising this question. We agree that inhibition of TICT is probably one of the reasons for the improved photostability. As stated in a paper from Liu's group (Chem. Soc. Rev., 2021, 50, 12656), TICT state is typically nonemissive and highly reactive. It was assumed that TICT state would promote the generation of free radicals, thus causing photobleaching. We found the addition of Tempo, a free radical trapping agent (Nature, 2019, 574, 516-521), does increase the photostability of Rhodamine B (Appendix Fig. 1), indicating the possibility to increase the photostability *via* inhibiting TICT formation. Meanwhile, DFT calculation indicates the lower inclination of **YL578** to form TICT state (Fig. 2h). Besides, it is worth noting that rhodamines with vibration structures also possess enhanced photostability (J. Am. Chem. Soc. 2018, 140, 7716). Thus, we assume that the superior photostability of YL578 could be the synergetic effect from the TICT inhibition and vibronic structure. The discussion was added in the revised manuscript (page 8, line 8-14).

Reviewer #2:

In this work, Jiang and coworkers mainly reported a new group of Rhodamine fluorescent dyes with improved brightness, photostability and Stokes Shift, which are revealed to be suitable for STED imaging in live cells. I think it is very interesting for the researchers in STED super resolution imaging, because a new STED fluorescent dye with excellent photo properties for live cell imaging and this kind of modifications on Rhodamine dye have not yet been reported before. Therefore, I think this work is novel and can be considered to be published in Nature communcaitions after addressing some concernings as follows:

1. *In the introduction section, the authors pointed out that an ideal fluorescent dye should satisfy some standards for STED imaging, high brightness, good photostability and different excitation and emission wavelength can be acceptable, the large Stokes Shift is also a beneficial point for STED, why?*

Response: We thank the reviewer's comment. In the previous review and research articles (Methods Appl. Fluoresc. 2015, 3, 042004; J. Am. Chem. Soc. 2017, 139, 12378), S. W. Hell et. al. have proved two main advantages of the dyes with large Stokes shift in STED imaging: (1) combine with probes with small Stokes shift to target different specimens and produce multicolor STED images with one depletion laser, such as three-color STED images in this work (Figure 4e), (2) avoid the re-excitation caused by the powerful STED light, which produces a diffuse halo around the sharp super-resolution image and photobleaching. We added and highlighted this point in the introduction of the revised manuscript (page 2, line 11-12).

2. *In Figure 1, the authors depicted a scheme that the modification from Rhodamine B to the target dyes, all parameters of the brightness, photostability and Stokes Shift are increased, which seems to be a contradiction. The fluorescent brightness and Photostability are increased, due to the introduction of the rigid scaffold into the molecule to reduce the energy loss, which are easy to be understood. However, the increase of the Stokes shift seems to be contradictory with the reduction of the energy loss, the authors should make more detailed discussion here.*

Response: We thank the reviewer for raising this question. As shown in Jablonski diagram, the energy of the excited fluorophores will be released through several processes, including internal conversion (E_{ic}), vibration relaxation (E_{vr}), intersystem crossing (E_{isc}), radiative transition (E_{rt}), and nonradiative transition (E_{nrt}) (Lakowicz J.R. Principles of fluorescence spectroscopy. New York: Springer, 2006). The Stokes shift is mainly determined by the vibrational relaxation in the excitation state (S_1 state). More energy loss in vibrational relaxation generates the enlarged Stokes shift. When the excited fluorophore relaxes to the ground state, the energy can be released in two ways: radiation transition (fluorescence) and non-radiative transition. The increased energy in the brightness could be from the decreased non-radiative transition. Therefore, we assumed that the introduction of 2-(2,2,2-trifluoroethyl)octahydropyrrolo[1,2-a]pyrazine group in **YL578** could intensify vibration relaxation in excitation state, enlarging the Stokes shift. Meanwhile, the inhibited TICT formation and the low environmental sensitivity of **YL578** reduce the energy loss through nonradiative transition (Fig. 2 and supplementary Fig. 10), which improves the brightness of dyes. Comprehensively, it is possible to produce fluorophore with higher brightness

and longer Stokes shift. The discussion was added in the revised manuscript (Page 8, line 1-4) and Supplementary Fig. 1a.

Reviewer #3:

In their manuscript titled “A synergistic strategy to develop photostable and bright dyes with long Stokes shift for super-resolution microscopy”, Jiang, Ren, and coworkers describe a strategy to increase the brightness, photostability, and Stokes shift of rhodamines and other fluorophore scaffolds through the incorporation of a bicyclic quinoxaline moiety in place of the standard N,N-dialkylamino group common to many fluorescent small molecules. The authors previously reported a series of quinoxaline fluorophores (rhodols in Chen et al. ACIE 2017, 56, 16611; rhodamines, pyronines, and oxazines in Ren et al. JACS 2018, 140, 7716) that displayed large Stokes shifts but poor quantum yields. The main structural modification here is the fusion of the second 3'-nitrogen substituent onto the quinoxaline to form an octahydropyrrolo[1,2-a]pyrazine, which—like the previous quinoxalines—increases the Stokes shift through the generation of vibronic structures. Although this ring system was reported in the 2017 ACIE paper, the authors show here that substituting the 2'-nitrogen with electron-withdrawing groups can rescue quantum yield (presumably by reducing TICT). The substituent best balancing brightness and Stokes shift (since EWGs do decrease the shift) is 2,2,2-trifluoroethyl. The resulting rhodamine “BDQF-6” shows improved QY and photostability compared to rhodamine B, surprising contrast as a self-labeling tag (HaloTag) ligand, and good behavior as a STED dye compared to two other rhodamines with similar spectra/wavelengths (see (3) below). Incorporation of the same ring system into rhodol, pyronine, coumarin, and Boranil dyes also significantly increases brightness and Stokes shift compared to the parent (N,N-dialkylamino) fluorophores.

The main advancement touted in this manuscript is interesting and potentially useful to the chemistry and microscopy communities. However, there are several key issues that should be addressed before this article can be considered for publication in Nature Communications:

- 1. The authors emphasize that the key structural modification (the octahydropyrrolo[1,2-a]pyrazine with electron-withdrawing groups) unites two strategies to improve fluorophores: vibronic structures and TICT inhibition. And it does that, to an extent. Why, though, is the “other” side of the rhodamines ignored? Why do all the rhodamines retain the poor, suboptimal N,N-diethyl group on the opposite side of the rhodamine? This moiety is a known liability from both a QY (TICT) and photostability (dealkylation) perspective. The authors should incorporate one or more of the known strategies to lessen TICT and improve brightness—several of which they themselves show in the abstract—such as the work of Foley (azabicycloheptane) or Lavis (azetidine). Replacing the N,N-diethyl with any number of better choices would retain the*

asymmetry of the rhodamines, provide better fluorophores for comparison to the state-of-the-art in rhodamine dyes, and demonstrate if all of these strategies are indeed synergistic.

Response: We thank the reviewer for pointing this out. We did not modify the other side of rhodamine since the DFT calculations showed that 2-(2,2,2-trifluoroethyl)octahydropyrrolo[1,2-a]pyrazine group (**S-6**) can inhibit the TICT formation of diethylamino group (Figure 2h). We agree with the reviewer's comments that inhibition of TICT formation could further improve the fluorophores, thus we integrated azetidine into **YL-578**, producing **YL-Az**. **YL-Az**, similar to **YL578**, displayed red-shifted absorption and emission spectra and longer Stokes shift than Rhodamine B. Interestingly, **YL-Az** and **YL-578** also showed similar photostability under the same irradiation conditions (Supplementary Fig. 13b-13d). This is consistent with our assumption that the introduction of 2-(2,2,2-trifluoroethyl)octahydropyrrolo[1,2-a]pyrazine moiety (S-6) on one side of rhodamines cannot only induce vibration feature in rhodamines, but also inhibit the TICT effect of the diethylamino group in the other side, resulting in a synergistic effect to improve the brightness, photostability, and Stokes shift. The synthesis processes and experimental results of **YL-Az** were added in the revised manuscript (Page 4, line 20-23) and supplementary information (Page 45, line 31-36; page 46, line 1-14).

- 2. Given work in the past 10-20 years, rhodamine B is far from the state-of-the-art as far as rhodamine brightness and photostability is concerned (and could be considered largely obsolete). There are now far brighter, cell-permeable rhodamines with quantum yields that approach 1. To convincingly demonstrate this strategy as a way to improve brightness and photostability, the authors should compare YL578 to more modern rhodamines such as Hell's tert-butyl rhodamines (Butkevich et al. JACS 2019, 141, 981) or Lavis's bis(azetidiny)rhodamines like JF549 (Grimm et al. NM 2015, 12, 244).*

Response: We are grateful to the reviewer for raising this important point. **YL578** was produced by replacing the diethylamino group with 2-(2,2,2-trifluoroethyl)octahydropyrrolo[1,2-a]pyrazine moiety in Rhodamine B. Thus, Rhodamine B was compared to confirm the strength of the new design strategy to improve brightness, photostability, and Stokes shift. As the reviewer suggested, the photophysical properties of **YL578** and JF549 were systematically compared and added in supplementary Fig. 10 and supplementary Fig. 13. **YL578** displayed the red-shifted absorption/emission spectrum and enlarged Stokes shift, while JF549 showed higher brightness. Interestingly, even **YL578** showed much better photostability than JF549 under the irradiation of 530 nm in EtOH (conditions used in J. Am. Chem. Soc. 2019, 141, 981–989), they possess similar performance in live-cell imaging. Considering the absorption/emission spectra and the depletion laser at 775 nm,

we compared the photostable probe 580CP-Halo, CPY-Halo, and JF608-Halo rather than JF549-Halo in STED imaging in the revised manuscript (Fig. 4a, b). The above experiments demonstrate the superior photostability of **YL578**, indicating that the new design strategy in this work would be a useful method to derive new dyes with high brightness, long Stokes shift, and outstanding photostability.

3.1 In demonstrating YL578-Halo as an effective STED dye, the authors compare it to “CPY-Halo” and JF608-Halo. Firstly, the structure of CYP-Halo looks to be incorrect in Supplementary Figure 3; the structure as drawn is actually tetramethylrhodamine-HaloTag ligand (TMR-Halo), but I assume it’s supposed to be the carbon analog (tetramethyl-carborhodamine). The reference cited for CYP-Halo is not correct (it does not describe CYP in any detail). How were CYP-Halo and JF608-Halo synthesized or obtained? It is not described in the methods.

Response: We thank the reviewer for these suggestions. The structure of CYP-Halo described in Supplementary Fig. 2 was corrected in the revised supplementary information. CPY-Halo (also called 610CP) and JF608-Halo are synthesized based on Hell’ work (Angew. Chem. Int. Ed. 2016, 55, 3290) and Lavis’s work (Nat. Methods 2017, 14, 987). Proper description and citation were added and highlighted in the revised manuscript (Page 5, line 36-39; page 9, line 14-15).

3.2 Beyond the similarity in emission maximum, it’s hard to see why these dyes were chosen. To my knowledge, JF608 has not been characterized as a “photostable dye in STED microscopy”. Given the significant difference in absorbance/excitation maxima for YL578 compared to CYP and JF608, should the imaging conditions not be adjusted?

Response: We thank the reviewer’s comments and suggestions. We agree with the reviewer that both the absorption and emission spectra of the probes are important in STED imaging. Thus, in the revised manuscript, we included not only CPY ($\lambda_{ex}/\lambda_{em} = 609/634$ nm) and JF608 ($\lambda_{ex}/\lambda_{em} = 608/631$ nm) sharing similar emission spectra of **YL578** ($\lambda_{ex}/\lambda_{em} = 578/634$ nm), but also 580CP ($\lambda_{ex}/\lambda_{em} = 582/607$ nm), which displays comparable absorption spectra with **YL578-Halo**. Meanwhile, to fairly compare the photostability of the above probes, the same imaging conditions (λ_{ex} 561 nm, detection range 570–700 nm, STED laser 775 nm) were applied in STED imaging. The imaging data of 580CP-Halo will be discussed in the following question. The sentence describing JF608 was also revised and highlighted in the revised manuscript (page 5, line 39-40).

3.3 Most importantly, how does YL578-Halo compare to the myriad STED

rhodamines and carbopyronines described by Hell over the past 10+ years?

Response: We thank the reviewer's comments and suggestions. CPY-Halo (also called 610CP) and 580CP-Halo reported by Hells' group (ACS Chem. Biol. 2018, 13, 475; Angew. Chem. Int. Ed. 2016, 55, 3290) are two of the most popular photostable probes in STED imaging. Meanwhile, since the replacement of dialkylamino in rhodamines with azetidino group has been utilized to improve photostability and brightness (Nat. Methods 2015, 12, 244; Nat. Methods 2017, 14, 987), JF608, the azetidino derivative of CPY-Halo, was also compared in STED imaging. Only 2-3 frames of STED images with >50% of the initial fluorescence intensity were obtained from CPY-Halo, 580CP-Halo, and JF608-Halo (Fig. 4a, 4b and Supplementary Fig. 21-23). In contrast, under identical conditions, **YL578-Halo** offers more than 9 frames of STED images with >50% of the initial fluorescence intensity while remaining the full width at half-maximum (fwhm) resolution of 57 ± 5 nm (Fig. 4a and 4b and Supplementary Fig. 24). Such experiments demonstrate the superior photostability of **YL578-Halo** in STED imaging. New experimental results of 580CP-Halo were added in the revised manuscript (Page 5, line 43-44) and supplementary Fig. 23.

3.4 A more advanced imaging experiment—such as 2-color STED (with a SiR dye, for example)—would also go a long way to demonstrating the utility of these dyes for complex studies.

Response: We thank the reviewer's suggestions. To demonstrate the utility of these dyes for complex studies, we performed multi-color (2-color and 3-color) live-cell STED imaging of vimentin filaments, microtubules, F-actins, and nuclear DNA by incubating **YL578-Halo**, Map555-actin, SiR-DNA, and GeR-Tub. Such images were added in the revised Figure 4e and Supplementary Fig. 28.

4. Although the Supporting Information is certainly extensive, the synthetic experimentals and chemical characterization data for many of the compounds are not currently of publishable quality. All experimentals should contain masses/volumes for all reagents, as well as a specific mass and yield for the product. Each compound should have a specific yield and "state" (yellow oil, white solid, etc.) rather than just a range of yields for each procedure. The salt forms (free base/inner salt? TFA salt?) and/or counterions (for the pyronine and rhodamine methyl ester) are also not currently specified. Most importantly, though, several of the compounds have NMR spectra that demonstrate inadequate purity (or are otherwise broad and uninformative), with (for the ¹H NMR) suspect integrations that exclude significant impurity peaks and (for the ¹³C NMR) cherry-picking of peaks (e.g., clusters of many smaller peaks with only 1 or 2 picked/reported). Some of the more concerning examples are S-3,

1, 5, YL578-Lyso, YL578-Halo, 10, 11, and 11-ALP. What is the reason for the CD₃OD/CDCl₃ solvent mixture used for many of the NMR spectra? Perhaps another solvent would resolve some of the broadness observed in many of the ¹H NMR? For compounds that do not have a ¹³C NMR, another assay of purity (e.g., HPLC trace) should be included.

Response: We are very grateful to the reviewer for raising these important points. The detailed information for the reagents, synthesis procedure, salted form, and counterions were added in the revised supplementary information. For some products, e.g. **11-ALP** (Appendix Fig. 2), different solvents (CDCl₃, CD₃OD, CD₃COCD₃) were used but none of them can dissolve these compounds well and provide satisfying NMR spectra. Thus, CD₃OD/CDCl₃ mixture was applied to perform NMR spectroscopy. As the reviewer suggested, we also tried different deuterated solvents to reduce the broadness in ¹H NMR of product **S-3** (Appendix Fig. 3-5). NMR spectra of the products (**S-3**, **1**, **5**, **YL578-Lyso**, **YL578-Halo**, **10**, **11**, and **11-ALP**) were measured again in the proper deuterated solvent. Meanwhile, the purity of all products without ¹³C NMR was confirmed by high-performance liquid chromatography (HPLC). We also added the crystal data of **YL578** in the revised supplementary information. We listed all the new NMR and HPLC spectra in the revised supplementary information.

5. *The authors curiously do not report a symmetrical rhodamine with the bis-CH₂CF₃ moiety (bis-S-6) on both sides. Although this dye would likely not display an enhanced Stokes shift due to symmetry, it would be an excellent way to showcase how this strategy specifically addresses TICT compared to existing approaches (i.e., by comparing the “bis-S-6 rhodamine” to Foley’s bis(azabicycloheptane)rhodamine, Lavis’s JF549, etc.).*

Response: As suggested by the reviewer, we have synthesized a symmetrical **S-6**-containing dye, **bis-YL**. Interestingly, **bis-YL** indeed possesses a higher quantum yield than Rhodamine B in EtOH probably due to the inhibited TICT effect. However, the brightness significantly reduced in PBS buffer solution because of aggregation-caused quenching (ACQ) indicated by the increased peak of H-aggregate at 550 nm. Notably, **Bis-YL** showed a smaller Stokes shift than **YL578**, indicating the role of asymmetry vibronic structure in enlarging the Stokes shift. The new data of **bis-YL** were added in Supplementary Fig. 10, Supplementary Fig. 13 and revised manuscript (page 4, line 23-26).

6. *It appears that many (if not all) of the enhanced Stokes shift, S-6-containing dyes have broader spectra than their parent (non-S-6) analogs. The authors should report FWHM, as the broadened spectra can certainly be a concern for multiplexed imaging.*

Response: We thank the reviewer's suggestion. The full width at half maximum (FWHM) was added in the revised supplementary information. The new probes indeed display broadened spectra than their parental dyes, while it is worth noting that the increased Stokes shift is favorable in multi-color imaging, such as two-color and three-color STED imaging in the revised Fig. 4e and supplementary Fig. 28.

7. *There does not appear to be a definition of the acronym BDQ/BDQF. What does this represent? If, as in their previous paper, the "DQ" is "decahydroquinoxaline", it should be changed, since these do not contain decahydroquinoxalines. They could be referred to as 1,2,3,4-tetrahydroquinoxalines (includes the rhodamine aryl ring but not the fused 5-membered ring) or octahydropyrrolo[1,2-a]pyrazines.*

Response: We thank the reviewer for pointing this out. The products in this work were renamed in the revised manuscript. The representative probes were renamed as **YL**, representing the YueLu district where we developed the dyes.

8. *How do the authors explain the surprisingly high contrast of YL578-Halo (18x)? The no-wash contrast of rhodamine ligands (most notably the SiR-type) is most often attributed to the open-closed equilibrium, where the dye is more closed (lactone, non-fluorescent) in solution but shifts to the open form (zwitterion, fluorescent) upon binding to protein. YL578 (YL578-Halo sans HaloTag ligand), however, has spectral properties consistent with a rhodamine that is quite open ($\epsilon = 88,800$) in solution. Moreover, the dioxane-water titration of YL578-Halo (Supplementary Figure 16c) shows essentially no dependence of fluorescence on dielectric constant (i.e., a high lactone-zwitterion equilibrium constant), and HaloTag binding of YL578-Halo elicits almost no change in fluorescence (Supplementary Figure 17).*

Response: We are grateful to reviewer 3 for pointing this out. We agree with the reviewer that the high contrast is probably not from the equilibrium shifting between non-fluorescent spirocyclic and fluorescent zwitterionic forms, considering the low D_{50} (Supplementary Figure 16c) and few changes of the absorbance of **YL578-Halo** in DMEM medium after 4 h incubation (Appendix Fig. 6). Recently, Chang's group reported that some transporters (solute carrier (SLC) transporters and ATP-binding cassette (ABC) transporters) have broad substrate specificity, which includes not only derivatives of endogenous natural substrates but also totally synthetic diverse imaging probes (Acc. Chem. Res. 2019, 52, 3097). Once the probes covalently bind to HaloTag, the contrast is mainly determined by the intracellular background signal from the unbound probes. We thus assume that the low background signal of **YL578-Halo** might be from the high efficiency of the transporters in HeLa cells to efflux the probes out. The discussion was added in the revised manuscript (Page 5, line 5-6).

9. There is an overwhelming number of Supplementary Figures and Tables (41 total) in the Supporting Information. Several of them are not referred to at all in the main text. The authors should find a way to condense/organize/pair down these figures to make the SI more navigable/readable. Why are Supplementary Table 1-4 separated into four discreet tables when they all show the same type of information for different dyes?

Response: We are grateful to the reviewer for pointing this out. The Supplementary Figures and Tables have been reorganized in the revised paper. Meanwhile, we referred to the ones that have been omitted at the highlighted positions in the main text.

10. "Till now" should be "Until now".

Response: "Till now" has been corrected to "Until now" in the revised manuscript (Page 3, line 5).

Appendix Fig. 1 Absorption maxima of RhB in the presence or absence of Tempo, a free radical trapping agent, were plotted as a function of irradiation time with a laser (1 W) at 530 nm. Solution concentrations were adjusted to be comparable to one another in terms of optical density at 530 nm.

EDQF-11-ALP-DMSO.1.fid

Appendix Fig. 2 ¹H NMR of 11-ALP in DMSO-d⁶.

Appendix Fig. 3 ¹H NMR of S-3 in CD₃OD.

Appendix Fig. 4 ^1H NMR of **S-3** in DMSO-d_6 .

Appendix Fig. 5 ^1H NMR spectrum of **S-3** in CDCl_3

Appendix Fig. 6 The absorbance of **YL578-Halo** with different concentration (5 (a), 10 (b), 20 (c), 50 (d) μM) in phenol red-free DMEM medium after 1.5 and 4 h incubation.

REVIEWER COMMENTS

Reviewer #1 (Remarks to the Author):

In the revised manuscript, authors have addressed my concerns, and it now is suitable for publication in the journal.

Reviewer #2 (Remarks to the Author):

The authors have well answered my concerns about the new Rhodamine derivatives used for STED imaging. I can accept their interpretation about the my questions. Therefore, i choose to recommend the work to be published in nature communicatons.

Reviewer #3 (Remarks to the Author):

The revision of the manuscript "A synergistic strategy to develop photostable and bright dyes with long Stokes shift for super-resolution microscopy" by Jiang, Ren, and coworkers offers a substantial expansion and improvement on the original submission. The additional dyes address most of my previous concerns, and the characterization data is (for the most part) significantly improved (although I wonder if, for some of the broad, uninformative NMR, a VT experiment could clean up any weird rotamer/conformational behavior). I would recommend this paper for publication in Nature Communications provided that the following points are addressed:

- (1) The experimental for YL-Az (Supp Scheme 20) is odd (and perhaps unprecedented if it is correct as written). Did the authors actually cross-couple a free rhodol (phenol) with azetidine? This would be unusual indeed. Or did they form the triflate, then do the cross-coupling and forget to include the triflation step? If so, the triflation needs to be added and the triflate intermediate characterized.
- (2) In some cases throughout the manuscript, the rhodamines are drawn as the iminium (nitrogen-centered cation), whereas in other cases, they are drawn as the oxygen-centered cation. For the sake of consistency and clarity, one form (ideally the more common iminium representation) should be used throughout the paper for all the rhodamines.
- (3) In the Fig 2d caption, "Absorption maxima" should be "Absorption at λ_{max} ".
- (4) In Fig 3j, "Probe 14" should be "YL578-Lyso"(?).
- (5) In Table 1 and Fig 5a, the use of the R1 and R2 generics is not, strictly speaking, correct. There is a single "Substituent (R)" column in the table which identifies either S-6 (which is a compound, not a substituent) or NEt₂ as "R", but doesn't say which (R1 or R2) they correspond to (obviously the quinoxaline spans R1 and R2, but that may not be understood by non-chemists). In Fig 5a, the "attachment" of R2 and R1 to the piperazine ring is confusing. I would rethink the way the generics are used. In Fig 5a, just drawing the two separate structures would probably be easier.

(6) In Supp Fig 13b, the caption should say "Absorption at λ_{max} " instead of "Absorption maxima". I'm also puzzled by Supp Fig 13b itself (the same data as Fig 2d but with the other three dyes added?)...how do the authors rationalize the performance of JF549 in 13b (~the same as RhB) with 13c/d (~the same as YL578 and YL-Az)?

(7) For Supp Fig 29, I find the inclusion of the spectra for the organic solvents to be somewhat unnecessary and cluttering. Since the focus of this paper is on the use of the dyes in advanced biological imaging, their properties in DCM, MeCN, etc. is of limited relevance (and the data is available in Supp Table 1 if a reader wants it). The plots would be much easier to read if they only included the abs and fl em for aqueous solution (PBS) and were labeled on the plot with the name of the dye. The authors could consider plotting the normalized abs and normalized fl em data on the same plot for all the dyes as in Fig 2c, which would reduce 20 graphs to only 10 and highlight the Stokes shifts of the dyes in a visual way.

(8) In the experimentals for cmpds 11 and 17, what is meant by perchlorate zwitterion? If the rhodol is a zwitterion (+/-), why would there be a counterion? Do they mean a perchlorate salt?

POINT-BY-POINT RESPONSE TO THE COMMENTS ON THE MANUSCRIPT “A SYNERGISTIC STRATEGY TO DEVELOP PHOTOSTABLE AND BRIGHT DYES WITH LONG STOKES SHIFT FOR SUPER-RESOLUTION MICROSCOPY”

We are very grateful to the reviewer for their insightful comments and suggestions in improving the quality of the paper. In the revised manuscript we have attempted to address all these points and changed the manuscript accordingly. Below we provide a point-by-point response to the comments of the reviewers.

Reviewer comments:

Reviewer #3:

- 1. The experimental for YL-Az (Supp Scheme 20) is odd (and perhaps unprecedented if it is correct as written). Did the authors actually cross-couple a free rhodol (phenol) with azetidione? This would be unusual indeed. Or did they form the triflate, then do the cross-coupling and forget to include the triflation step? If so, the triflation needs to be added and the triflate intermediate characterized.*

Response: We would like to thank the reviewer for raising this question. **YL-Az** was synthesized through the triflation step and we have added the characterized triflate intermediate in the supplementary Scheme 20 (page 45, line 4-23).

- 2. In some cases throughout the manuscript, the rhodamines are drawn as the iminium (nitrogen-centered cation), whereas in other cases, they are drawn as the oxygen-centered cation. For the sake of consistency and clarity, one form (ideally the more common iminium representation) should be used throughout the paper for all the rhodamines.*

Response: As suggested by the reviewer, we have adjusted the rhodamines to the iminium in the revised manuscript and supplementary information.

- 3. In the Fig 2d caption, "Absorption maxima" should be "Absorption at λ_{max} "*

Response: We thank the reviewer for pointing this out. "Absorption maxima" has been changed to "Absorption at λ_{max} " in the Fig. 2d caption.

- 4. In Fig 3j, "Probe 14" should be "YL578-Lyso"(?).*

Response: We are grateful to the reviewer for raising this question. "Probe 14" has been changed to "YL578-Lyso" in the revised Fig. 3j.

5. In Table 1 and Fig 5a, the use of the R1 and R2 generics is not, strictly speaking, correct. There is a single "Substituent (R)" column in the table which identifies either S-6 (which is a compound, not a substituent) or NEt₂ as "R", but doesn't say which (R1 or R2) they correspond to (obviously the quinoxaline spans R1 and R2, but that may not be understood by non-chemists). In Fig 5a, the "attachment" of R2 and R1 to the piperazine ring is confusing. I would rethink the way the generics are used. In Fig 5a, just drawing the two separate structures would probably be easier.

Response: We are grateful to the reviewer for raising this important point. To make it easier to read, we have listed the structural formulas of all dyes in table 1 and Fig. 5a.

6. In Supp Fig 13b, the caption should say "Absorption at λ_{max} " instead of "Absorption maxima". I'm also puzzled by Supp Fig 13b itself (the same data as Fig 2d but with the other three dyes added?). How do the authors rationalize the performance of JF549 in 13b (~the same as RhB) with 13c/d (~the same as YL578 and YL-Az)?

Response: We thank the reviewer's comment. "Absorption maxima" has been changed to "Absorption at λ_{max} " in the revised supplementary Fig. 13b. We only showed YL578 and Rhodamine B in Fig. 2d to indicate the strength of new design strategy for improving the dye's photostability. In Supplementary Fig 13, we tested again the photostability of the five dyes under the same conditions utilized in Fig. 2d as the reviewer suggested.

To reduce the interference caused by the different solubility of the dyes in aqueous solution, we chose ethanol containing 0.1% TFA as the solvent to test the photostability, following the conditions in the previous work of Hell's group (J. Am. Chem. Soc. 2019, 141, 981). Interestingly, we found JF549 displayed poor photostability in EtOH while excellent performance in cellular imaging, which was assumed to be resulted from the difference between organic solvent and intracellular environment. The discussion was added in the revised Supplementary Fig 13 (page 10, line 12-13).

7. For Supp Fig 29, I find the inclusion of the spectra for the organic solvents to be somewhat unnecessary and cluttering. Since the focus of this paper is on the use of the dyes in advanced biological imaging, their properties in DCM, MeCN, etc. is of limited relevance (and the data is available in Supp Table 1 if a reader wants it). The plots would be much easier to read if they only included the abs and fl em for aqueous solution (PBS) and were labeled on the plot with the name of the dye. The authors could consider plotting the normalized abs and normalized fl em data on the same plot for all the dyes as in Fig 2c, which would reduce 20 graphs to only 10 and highlight the Stokes shifts of the dyes in a visual way.

Response: We'd like to thank the reviewer for pointing this out. We also agree with the reviewer that the normalized absorption and emission spectra only for

PBS buffer solution in supplementary Fig. 29 would be easier to read. The figure was changed accordingly and added in the revised supplementary information.

8. *In the experimentals for compounds 11 and 17, what is meant by perchlorate zwitterion? If the rhodol is a zwitterion (+/-), why would there be a counterion? Do they mean a perchlorate salt?*

Response: We thank the reviewer for raising this question. The description of compounds 11 and 17 have been corrected in the revised supplementary information (page 33, line 32; page 34, line 7).